# Early childhood risk and protective factors and their association with adolescent sexual behaviors: A Latent Class Analysis

Qingyang Liu [1,2]*, Sara A. Vasilenko[2], Xiafei Wang[3], Rachel A. Razza[2]

1 Department of Psychology, Arizona State University, Tempe, Arizona, United States of America,
2 Department of Human Development and Family Science, Syracuse University, Syracuse, New York,
United States of America, 3 College of Social Work, University of Kentucky, Lexington, Kentucky, United
States of America

* qingyangliu96@gmail.com

## Abstract

### Background

Prior research examined associations between social-ecological risk and protective factors and adolescents' sexual behaviors. However, scarce research explored how factors in early childhood impact adolescents' sexual behavior.

### Methods

In this study, we used a socioeconomically and racially diverse sample from the Future of Families and Child Wellbeing Study ($N = 3,281$; 51.3% male; 21.76% White, 50.1% Black, 24.61% Latinx) to uncover latent classes of early childhood social-ecological risk and protective factors, examine the demographic characteristics of each latent class, and examine how these classes are associated with adolescents' sexual behaviors.

### Results

Four latent classes were identified: Multidimensional Protective (21.5%), Socioeconomic Status Risk (59.2%), Neighborhood Risk (7.3%), and Low Father Education (12%). Compared to the Multidimensional Protective class, adolescents in the Socioeconomic Status Risk class had a higher prevalence of sexual initiation, early sexual initiation, sex without a condom at first intercourse, and multiple sexual partners. Adolescents in the Socioeconomic Status Risk class had a higher prevalence of having sex without a condom at first intercourse than those in the Neighborhood Risk class. Adolescents in the Low Father Education class had a higher prevalence of having multiple sexual partners than those in the Multidimensional Protective class.

**Data availability statement:** The de-identified dataset used in this study is publicly available on Figshare: Liu, Qingyang (2025). FFCW_FilterCases_NOLabels.csv. figshare. Dataset. https://doi.org/10.6084/m9.figshare.29619275.v1. The code used to conduct the latent class analysis is included in the Supplemental Material.

**Funding:** This research was supported by the grant R03 HD096101 from the Eunice Kennedy Shriver National Institute of Child Health and Human Development and a David B. Falk College of Sport and Human Dynamics Tenure-track Assistant Professor Research Seed Grant to Sara Vasilenko and Xiaofei Wang. The content is solely the responsibility of the authors and does not necessarily represent the official views of NICHD or the National Institutes of Health.

**Competing interests:** The authors have declared that no competing interests exist.

## Conclusion

These findings suggest the need for early, targeted interventions, such as parenting support programs, father engagement initiatives, and community-based efforts to strengthen neighborhood cohesion, to promote long-term sexual health during adolescence.

---

Adolescence marks a crucial phase in sexual development; understanding the predictors of adolescent sexual behavior is important from both normative developmental and health risk perspectives [1,2]. About 60% of individuals engage in their first vaginal intercourse by age 18, with an average age of first intercourse of 17 among adolescents in the U.S. (National Center for Health [3]). Engaging in sexual activity at an early age, often defined as before age 16, is associated with adverse physical and mental health outcomes, including elevated risk of sexually transmitted infections (STIs) and increased depressive symptoms [4,5]. Additionally, unprotected sex and having multiple sexual partners heighten risks for STIs and unintended pregnancy [6]. Notably, engaging in unprotected sex at first sexual intercourse is linked to subsequently risky sexual behaviors, including unintended pregnancy, a greater number of lifetime partners, and increased rates of STIs [7,8].

A comprehensive understanding of early childhood risk and protective factors influencing adolescent sexual behaviors helps to design early intervention programs that specifically target individuals with past histories that may place them at greater risk. Although longitudinal studies have examined early childhood factors, such as maternal warmth [9], children's self-regulation [10], and children's behavioral problems [11,12], in relation to adolescent sexual behaviors, most studies have adopted a variable-centered approach, focusing on the influence of individual predictors. Few studies have used a person-centered approach to explore how constellations of early childhood social-ecological factors shape adolescents' sexual behaviors. This approach allows researchers to identify subgroups of children with distinct patterns of early risk and protective factors and recognize which children may be at greatest risk for engaging in adolescent sexual risk behaviors. In this study, we applied a person-centered approach among a socioeconomically and racially diverse sample to examine the underlying subgroups of risk and protective factors in early childhood and their associations with adolescent sexual behaviors.

### Theoretical framework

Several prominent theories that discuss the critical roles of childhood adversity, as well as the interaction of multiple risk and protective factors, guide this study [13,14,15]. First, life history theory underscores the importance of early childhood and the long-term impact of early life experiences on adolescent sexual behaviors [13]. Derived from the evolutionary perspective, this theory posits that humans make trade-offs regarding reproduction because of limited time and environmental resources [16].

Individuals make optimal trade-offs to make reproductive decisions that are most adaptive to their living environments. The first five years of life experience critically shape individuals' perceptions of their environmental resources. These perceptions influence individuals' decision-making regarding the allocation of limited environmental resources later in life [13]. For example, children raised in a risky and stressful environment, such as family dysfunction and poverty, may prioritize reproductive success by developing a *fast* reproductive strategy, including early sexual onset, more sexual partners, and offspring. Conversely, children reared in a positive and supportive childhood environment characterized by parental warmth are likely to develop a *slow* reproductive choice that prioritizes personal growth and the acquisition of knowledge and skills over reproduction at an early age [17]. Furthermore, the social-ecological framework provides a more comprehensive understanding of diverse, co-occurring risk and protective factors that could influence adolescents' sexual behaviors. Bronfenbrenner underscores that individual development is shaped by interconnected and dynamic social settings, such as the individual, family, and neighborhood, where individuals live and interact [14]. Human development, including adolescent sexual behavior, is shaped by the interaction within and between these different developmental contexts.

Additionally, Masten's risk and protective framework highlights how outcomes like adolescent sexual behaviors may be influenced by multidimensional contextual factors [15]. In this framework, risk factors refer to factors that increase the likelihood of experiencing negative outcomes, while protective factors refer to factors that buffer the negativity and increase the probability of positive outcomes. Experiencing protective factors under adverse risky contexts can protect against negative impacts and promote resilience and positive outcomes [15]. For instance, living in neighborhoods with greater collective efficacy could buffer the negativity of environmental violence and promote children's social and emotional development [18]. Examining the interrelatedness between risk and protective factors could allow researchers to examine how protective factors can buffer negative influences on individuals' development.

While these theoretical frameworks provide a justification for examining various early risk and protective factors, they do not specify which particular factors are most relevant. Guided by life history theory, we selected predictors from early childhood. Grounded in the social-ecological framework, we focus on factors at the individual level (i.e., children's self-regulation) and factors within their principal interpersonal environments, including parent-level factors (i.e., parental impulsivity, maternal warmth), familial-level contexts (i.e., family socioeconomic status), and community-level contexts (i.e., neighborhood collective efficacy) that may interactively impact adolescent sexual behaviors. Informed by the risk and protective framework, we incorporated indicators that provide insights into both risk and protective elements. In the subsequent sections, we provide theoretical and empirical support for the inclusion of each of the risk and protective factors in this study.

## Children's self-regulation and adolescent sexual behaviors

At the individual level, early self-regulation serves as a protective factor for later adaptive behaviors. Some scholars also use the term self-control interchangeably to capture self-regulation [19], which represents the ability to control, monitor, and modulate attention, emotions, and behaviors to meet environmental demands [20]. Based on self-control theory, individuals with strong self-regulation are better able to resist immediate temptations and are less likely to engage in impulsive and risky behaviors [21]. Individual differences in early self-regulation have enduring effects on adolescents' behavior and well-being [10]. Indeed, self-regulation has been identified as a key protective factor for reducing adolescent risky behaviors [22,23]. For instance, higher levels of self-regulation in middle childhood were associated with a lower likelihood of having multiple sexual partners [24] and reduced engagement in early sexual initiation and unprotected sex [25]. Similar patterns emerge in adolescence, such that higher self-regulation was linked to delayed sexual onset, a lower likelihood of having multiple sexual partners, and more consistent contraceptive use [25–27,23].

## Parental impulsivity and adolescent sexual behaviors

Intergenerational transmission theory suggests that parental impulsivity can be passed down across generations, subsequently influencing children's behaviors, including diminished levels of self-regulation [28]. Parental impulsivity is

characterized by taking action without thorough consideration of the consequences. Parents with higher impulsivity could adopt harsh parenting and inconsistent discipline, which limits children's involvement in decision-making [29]. Children raised by impulsive parents are more likely to develop impulsive decision-making during adolescence, rendering them vulnerable to risky behaviors, such as multiple sexual partners, unprotected sex, and substance use [30]. Research also suggests that adolescents' impulsivity is a risk factor for adolescent sexual risk-taking, including multiple sexual partners, unplanned pregnancy, and a history of STIs [31,32]. Thus, at the parent level, parental impulsivity could serve as an early risk factor for risky adolescent sexual behavior via both intergenerational transmission and its negative influences on parenting behaviors.

## Maternal warmth and adolescent sexual behaviors

Based on attachment theory, maternal warmth is vital in fostering secure attachment [33]. When mothers exhibit warmth, support, and responsiveness to their children's needs, children receive positive cues from their caregivers and are more likely to develop a secure attachment with their mothers and caregivers. Consequently, maternal warmth contributes to the internalization of positive values to support children's decision-making and impact adolescents' establishment of boundaries that influence their engagement in sexual behaviors [34,35]. Higher levels of maternal warmth are associated with postponed sexual onset [36] and fewer sexual partners among adolescents [37]. Furthermore, early maternal warmth is linked to a lower likelihood of sexual intercourse in adolescent boys [9]. Together, these findings highlight how maternal warmth serves as a protective factor, contributing to the delay and reduced risk of adolescents' sexual activities.

## Socioeconomic status (SES) and adolescent sexual behaviors

At the family level, SES typically includes parental educational levels and household income [38]. Lower household income often co-occurs with other risk factors, such as maltreatment, low parental education, and unemployment, which are risk factors for early teen pregnancy [39] and multiple sexual partners [40]. Another component of family SES, parental education, serves as a proxy for parenting practices within the home environment to support children's development, such as providing learning materials, establishing structure, and creating less chaotic living conditions [41]. Parents with lower levels of education may face job strain, limited time to supervise children's behaviors, and higher parenting stress, contributing to reduced monitoring and communication [42].

While few studies have directly examined early childhood SES and later sexual behaviors, existing research shows that lower SES in adolescence correlates with adolescent sexual behaviors, including the increased likelihood of engaging in sexual intercourse [43,44]. Notably, these associations may vary depending on which SES component is examined. For example, maternal education appears more consistently linked to adolescent sexual initiation than household income [45]. However, most studies examined SES in isolation instead of considering other influential contexts, such as children's behaviors, parenting, and neighborhood, that interact with SES and collectively impact adolescents' sexual behaviors. A more integrative approach that considers co-occurring contextual factors could better identify early mechanisms and factors that mitigate the risks associated with lower SES.

## Neighborhood collective efficacy and adolescent sexual behaviors

At the community level, neighborhood environment significantly influences early child development by shaping parenting dynamics and parent-child relationships, which subsequently impact adolescent sexual behaviors [14]. Social control theory posits that strong social bonds and effective informal controls within neighborhoods could buffer against deviant or risky behaviors [46]. Sampson [47] introduced neighborhood collective efficacy as a key process linked to positive outcomes, comprising social cohesion, in which residents know and help each other, and informal social control, in which residents are willing to intervene when needed [47]. Research showed that higher levels of neighborhood collective efficacy in adolescence are associated with delayed sexual initiation [48] and fewer sexual partners [49].

Although studies have not directly examined early childhood neighborhood collective efficacy in relation to adolescent sexual behaviors, evidence has found that early childhood neighborhood collective efficacy is associated with lower behavioral problems, better social skills, and fewer mental health problems in adolescence [50,51]. Moreover, collective efficacy may complement parenting efforts by alleviating parenting stress. Parents in high-efficacy neighborhoods benefit from stronger social networks and a greater sense of safety, which are associated with lower risks of child maltreatment [52,53]. Thus, early neighborhood collective efficacy could serve as a protective factor against adolescent risky sexual behaviors by fostering both safer communities and more supportive family environments.

## Personed-centered approach

Although extant literature has linked factors at the individual (i.e., self-regulation), parental (i.e., parental impulsivity and parental warmth), familial (i.e., SES), and community (i.e., neighborhood collective efficacy) levels to adolescents' risky sexual behaviors, little research has examined how these contextual factors co-occur or interact. While previous studies applied the social-ecological framework to examine risk and protective factors for sexual behaviors across multiple levels [54] and to understand behavioral problems that covary with early sexual risk-taking, such as adolescent delinquency [55] and violent behaviors [56], these studies largely adopted a variable-centered approach, which examined these factors in isolation. However, the variable-centered approach may overlook the complex and interactive nature of contextual influences. In contrast, a person-centered approach aims to uncover underlying subgroups of individuals with shared patterns of risk and protective factors [57]. This approach enables a more comprehensive understanding of how protective factors can mitigate the adverse effects of risk factors and help recognize the synergistic and interconnected nature of early factors contributing to adolescents' sexual behaviors.

We also included demographic characteristics as predictors of latent class membership and controls in the outcome model. Prior research has shown that gender differences in sexual behaviors exist, such that male adolescents are more likely to engage in early sexual intercourse than female adolescents [58]. Children's age was related to the development of early self-regulation [20]. Parental age was related to parenting strategies, such that older parents were more likely to adopt warm and supportive parenting styles [59]. Living with married or two-parent families has been linked to delayed sexual initiation [36]. Racial/ethnic differences have also been observed, with Black and Latinx individuals more likely to experience neighborhood risk, have multiple sexual partners, and reduced contraceptive use [60,61,62].

## Current study

Prior research has documented various social-ecological risk and protective factors linked to adolescents' risky sexual behaviors. To comprehensively capture the multidimensionality and co-occurrence of these factors, we employed Latent Class Analysis (LCA) to identify heterogeneous classes marked by the risk and protective factors in early childhood and their long-term association with adolescents' sexual behaviors. We formulated three research questions and hypotheses: (1) What distinct latent classes of social-ecological risk and protective factors are present at age 5? Although the person-centered approach selects the best-fitting model based on the data without presupposing specific hypotheses, based on prior literature, we anticipate identifying latent classes demonstrating an absence of risk factors/high prevalence of protective factors, classes with domain-specific risks (e.g., socioeconomic risk), and classes experiencing a mix of specific risk and protective factors. (2) Which child and family demographic characteristics (i.e., child sex, child age, parental age, marital status, race/ethnicity, and adolescents' sexual attraction) predict class membership? We hypothesized that being a racial/ethnic minority, living in a single-parent or unmarried household, and being male would be associated with being in classes marked by higher risk and fewer protective factors. (3) How is class membership associated with adolescent sexual behaviors (i.e., sexual initiation, early sexual initiation, sex without a condom at first intercourse, and multiple sexual partners), controlling for demographic factors? We expected that adolescents in classes marked by higher risk factors would be more likely to engage in risky sexual behaviors compared to those in classes with more protective factors.

## Method

### Participants and procedures

The current study used secondary data from the Future of Families and Child Wellbeing Study (FFCWS), a longitudinal study that followed a birth cohort of 4,898 children born from 1998 to 2000 across 20 large urban cities in U.S. By design, the cities were selected to represent all U.S. cities with populations of over 200,000 and purposefully oversampled the children born to unmarried parents ($n = 3,712$ vs. $n = 1,186$ children born to married parents; [63]. At baseline, mothers were recruited at the hospitals and completed the initial interviews within 48 hours of their child's birth. Fathers were interviewed soon after that. Researchers conducted follow-up phone interviews, home interviews with mothers and fathers, home observations, and in-home direct child assessments at ages 1, 3, 5, 9, 15, and 22. At age 15, adolescents completed self-report surveys either at home or through phone interviews. The data from the current study were drawn from children ages 5 and 15 from the home interview, the mother core survey, the father core survey, and the teen survey. For the analytic sample, we included individuals with at least one valid data point on sexual behaviors at the year 15 assessment and year 5 indicators, resulting in a final analytic sample of 3,281 (see Fig 1 for the sample selection process). FFCWS data collection procedures were approved by the Institutional Review Board (IRB) at Columbia University and Princeton University. As this study involved secondary data analysis, it was exempt from further IRB review by the authors' institution.

### Measures

After running preliminary analyses, we decided to dichotomize the ordinal/continuous scales (details of descriptive statistics for original continuous variables were provided in S1 Table). We first ran a Latent Profile Analysis using originally continuous indicators, but found that the results were not interpretable (S2 Table). Specifically, while there were small differences in the level of particular indicators across the different classes, their interpretation was substantively the same (e.g., high vs. slightly higher). These classes did not yield classes that were qualitatively different from each other and thus did not achieve our goals of demonstrating the different types of individual risk environments. Thus, we decided to dichotomize indicators based on the mean as the cut-off point into the binary level, in which 1 *(higher/lower risk of that specific indicator)* and 0 *(lower/more risk of that specific indicator)*. While there are limitations to this categorization, we

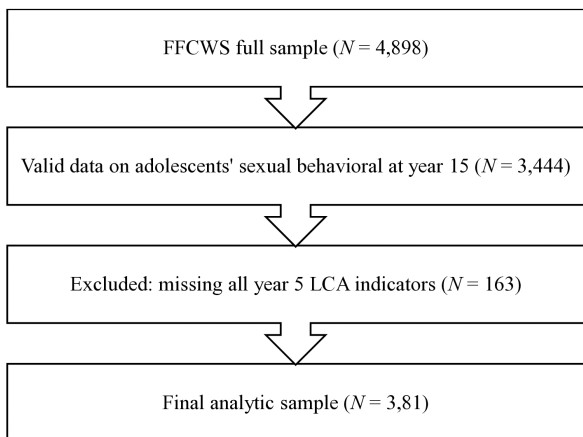

Note. FFCWS = Future of Families and Child Wellbeing Study. Participants included in the analytic sample had valid data on sexual behaviors at Year 15 and at least one non-missing value on Year 5 indicators.

**Fig 1. Flowchart of the Analytic Sample Selection Process.**

used it in order to uncover interpretable classes in line with our research aims (see [64], for further discussion of the rationale for dichotomizing indicators).

### Children's self-regulation

Children's self-regulation was measured with 25 items from the maternal report on the Child Behavioral Checklist in year 5 [65]. The 25 items have been used as a composite measure of self-regulation in prior literature (e.g., [66]). Mothers were asked to rate the frequency of children's behavior on a three-point Likert scale as 0 *(not true)*, 1 *(somewhat or sometimes true),* and 2 *(very true or often true).* An example item included "Child is impulsive or acts without thinking." Originally, the higher scores indicated a lack of self-regulation. In the current analyses, we reverse-coded all 25 items, with a high score indicating better self-regulation. Then, we created a sum score of all 25 items (Omega = .88) and then dichotomized the variable by a mean split, with values higher than the mean coded as 1 *(high in self-regulation)* and values lower than the mean coded as 0 *(low in self-regulation).*

### Maternal warmth

Maternal warmth was assessed by the interviewer's ratings of observation of mother-child interaction from the Home Observation for Measurement of the Environment (HOME) during the in-home visits in year 5 [67]. Eight dichotomous items were used from the HOME scales to reflect the interviewer's observation of the mother's affection and responsiveness toward the child during the in-home visit. An example item was "mother's voice conveys positive feelings when speaking of or to the child." We created a sum score of all 25 items (Omega = .84) and then dichotomized the variable using a mean split, with values higher than the mean coded as 1 *(high in maternal warmth)* and values lower than the mean coded as 0 *(low in maternal warmth).*

### Neighborhood collective efficacy

In year 5, mothers indicated their perception of the neighborhood collective efficacy through two widely accepted and slightly modified 10-item subscales from the Informal Social Control Scale and Social Cohesion and Trust Scale [47]. The Omega scores for the two subscales were.90 and.82, respectively. The Informal Social Control Subscales asked mothers to indicate how likely the neighbors would intervene or get involved in a series of scenarios occurring in the neighborhood. An example item included "children were skipping school and hanging out on the street." Items were originally coded on a 4-point Likert scale as 1 *(very likely)*, 2 *(somewhat likely)*, 3 *(not very likely)*, and 4 *(very unlikely)*. All five items in this subscale were reverse-coded, with higher scores indicating higher levels of neighborhood collective efficacy. The Social Cohesion and Trust Scale asked mothers to report to the extent that they agree with the following statements, such as "this is a close-knit neighborhood that is cohesive or unified." Each of the items was originally coded on a 4-point Likert scale, including 1 *(strongly agree)*, 2 *(agree)*, 3 *(disagree)*, and 4 *(strongly disagree)*. Higher scores indicated higher levels of neighborhood collective efficacy. After negatively worded statements were reverse-coded, we created a sum score of all 10 items (Castillo et al., 2020; Ma et al., 2018) and then dichotomized the variable by a mean split, with values higher than the mean were coded as 1 *(high in neighborhood collective efficacy)*, and values lower than the mean were coded as 0 *(low in neighborhood collective efficacy).*

### Poverty status

Considering FFCWS purposefully oversampled unmarried mothers who did not live with the child's father, the current study aimed to distinguish differences in household poverty status by including both parents' reports on household income. Therefore, we included indicators of maternal poverty status and paternal poverty status. Originally, FFCWS constructed the poverty status as five categories, which 1 *(0–49% of the federal poverty threshold)*, 2 *(50–99% of the*

*federal poverty threshold)*, 3 *(100–199% of the federal poverty threshold)*, 4 *(200–299% of the federal poverty threshold)*, and 5 *(300%+ of the federal poverty threshold)*. The first three categories were combined and coded to reflect the 0 *(poor group)*, and the fourth and fifth categories were combined and coded to indicate the 1 *(non-poor group)* to reflect mothers' and fathers' poverty status separately, following the coding approach used in prior literature [68].

### Parental educational attainment

Parental educational attainment was measured separately in two indicators in year 5 to assess mothers' and fathers' self-report of their highest levels of education attained. FFCWS coded mothers' and fathers' education into four categories: 1 *(less than high school)*, 2 *(high school or equivalent)*, 3 *(some college or technical)*, and 4 *(college or graduate school)*. The first two categories were combined and coded as 0 *(high school or less)*, and the third and fourth categories were combined and coded as 1 *(college or higher)* to measure mothers' and fathers' educational attainment separately.

### Parental impulsivity

In year 5, the mother and father were asked about dysfunctional impulsivity behaviors as a measure of the capacity for controlling impulsive behaviors. Parental impulsivity was measured by two items from Dickman's impulsivity scale [69]: "I often get into trouble because I do not think before I act," and "I often say and do things without considering the consequences." Each of the items was originally coded on a 4-point Likert scale, including 1 *(strongly agree)*, 2 *(agree)*, 3 *(disagree)*, and 4 *(strongly disagree)*. We created a sum score (Omega = .79) and dichotomized the variable by mean-split, with values higher than the mean coded as 1 *(low in impulsivity)* and values lower than the mean coded as 0 *(high in impulsivity)* for mother and father, respectively.

### Correlates of class membership

Seven variables were included as predictors of latent class membership and served as control variables in the model when using early childhood protective and risk classes to predict sexual behaviors. Child sex was the mother's self-report at birth, which was 0 *(boy)* and 1 *(girl)*. Child age was the mother's self-report in year 5 as a continuous score in months. The mother's age was the mother's self-report in year 5 as a continuous score in years. The father's age was the father's self-report in year 5 as a continuous score in years. Marital status was the mothers' self-report of their marital status at the child's birth and was coded as 0 *(single/in a relationship)* and 1 *(married)*. Mother's race was self-reported as 1 *(White)*, 2 *(Black)*, 3 *(Latinx)*, and 4 *(Other/Mixed race)*. Adolescents' sexual attraction was self-reported as 1 *(same-sex attraction)* and 0 *(do not have same-sex attraction)*.

### Outcomes: Adolescent sexual behaviors at year 15

Four measures reflected diverse facets of adolescent sexual behaviors: sexual initiation, early sexual initiation, sex without a condom at first intercourse, and multiple sexual partners by year 15. *Sexual initiation* assessed whether or not adolescents self-reported ever engaging in sexual intercourse at the year 15 survey and was derived from two items. The first item asked adolescents who had self-reported having a romantic relationship whether or not they had sexual intercourse with the current partner. The second item asked adolescents who did not report a current partner or did not have sexual intercourse with this romantic partner whether they had sexual intercourse with anyone in their lifetime. Adolescents' responses were coded as 0 *(no, never had sexual intercourse)* and 1 *(yes, had sexual intercourse)* if they reported sexual intercourse in either of these two items. Note that although the items referred specifically to "sexual intercourse," we included all adolescents in the study, regardless of their sexual orientation, as individuals with a same-sex attraction often engage in sexual risk behaviors, including engaging in vaginal intercourse, and can be at greater risk of negative outcomes including pregnancy involvement [70].

*Early sexual initiation* was measured by whether or not adolescents self-reported first sexual intercourse before age 16 from one item, which asked how old adolescents were when they had sex for the first time. In the year 15 data, adolescents' ages ranged from 14 to 19 years. Thus, we used age 16 to reflect a definition of early sexual intercourse consistent with prior research that defined early sexual intercourse as occurring before age 16 [5]. We dichotomized this variable by the age 16 cut-off point, in which items with values of age 16 or higher were coded as 0 *(do not have early sexual initiation)*, and values with age 15 and lower were coded as 1 *(early sexual initiation)*.

*Sex without a condom at first intercourse* was measured by one item that asked whether or not adolescents used a condom with their partner the first time they had sex. The item was originally coded as 1 *(yes)* and 2 *(no)*. We recoded the response as 0 *(no sex/sex with a condom)*, which included individuals who reported sex with a condom and individuals who did not engage in sexual intercourse during their lifetime, and 1 *(sex without a condom)*, consistent with coding approach in prior studies [71,72].

*Having multiple sexual partners* was measured by one item that asked about the total number of individuals adolescents have had sex within their lifetime. We dichotomized this variable, with values greater than or equal to two coded as 1 *(multiple sexual partners)* and values lower than two or those who had never engaged in sexual intercourse coded as 0 *(did not have multiple sex partners)*.

## Data analysis

The descriptives, correlation, attrition analyses, and missing data analyses were examined in R Studio 4.2. Latent Class Analysis [73] was used to uncover the underlying class membership based on the response patterns of early childhood risk and protective factors. The analysis proceeded by using the Bolck-Croon-Hagenaars (BCH) three-step estimation approach [74,73]. The BCH three-step uses logits to fix the class membership and to prevent the class membership from shifting after adding the predictors and distal outcome into the model, and provides less biased estimates than classify-analyze approaches. In the first step, we estimated the class enumeration process of latent class models with 1−6 classes, and selected the optimal number of latent classes based on both fit statistics and interpretability. Several fit statistics were used to evaluate the optimal class number to retain, including Akaike's Information Criterion (AIC), Bayesian Information Criterion (BIC), Consistent Akaike Information Criterion (CAIC), and Vuong-Lo-Mendell-Rubin adjusted likelihood ratio test (VLMR-LRT; [75]). VLMR-LRT tested if the K-1 classes fit the data significantly worse than the K-class model. To test our second research aim, after participants were assigned a probability of membership in each class, we used a latent class probability weighted multinomial regression model to examine demographic predictors (i.e., child sex, child age, mother age, father age, marital status, mother's race, and adolescents' same-sex attraction) to the prevalence of membership in each class to investigate how each predictor is associated with class membership compared to the reference group. In the third step, we conducted analyses of how latent class membership was associated with adolescents' sexual behaviors in year 15 using the BCH approach while controlling for demographic covariates. Full-information Maximum Likelihood (FIML) was used to handle missing data, which allows the model to preserve all available data and produces less biased estimates than deleting cases or imputing the sample mean [76]. FIML estimation with the robust maximum likelihood to handle nonnormality and missing data in Mplus 8.7. Data syntaxes were provided in S1 File.

## Results

### Descriptive statistics and missing data analysis

The demographic characteristics of the analytic sample were 51.3% male. Mothers reported children's age in year 5 ranging from 57 to 72 months ($M_{childage5} = 61.63$ months, $SD = 2.74$). Mothers' race was 21.76% White, 50.1% Black, 24.61% Latinx, and 3.52% Other/Mixed race; 75.77% of mothers were single compared to 24.23% married at the child's birth. At year 5, mothers' self-reported age ranged from 20 to 50 years ($M_{motherage5} = 30.23$ years, $SD = 6.02$), and fathers'

self-reported age ranged from 20 to 53 years ($M_{fatherage5}$ = 32.1 years, $SD$ = 7.12). At year 15, adolescents self-reported age ranging from 14 to 19 years ($M_{youthage5}$ = 15.59 years, $SD$ = .77). Adolescents self-reported same-sex attraction (9.15%) in year 15. Table 1 displays the descriptive statistics, including the number of full cases, sample mean, standard deviation, and frequencies of early childhood risk and protective factors, as well as adolescent sexual outcomes. Table 2 presents the bivariate correlations among LCA indicators, adolescent sexual outcomes, and demographic covariates.

Little's Missing Completely at Random (MCAR) test among the indicators before dichotomizing was not significant, suggesting that the data were missing completely at random ($\chi^2$ (13,031) = 11,861, $p$ > .05). We ran attrition analyses to examine the analytic sample ($n$ = 3,281) compared to individuals who were removed from the original sample ($n$ = 1,617) for the demographic variables and early childhood indicators in year 5. Regarding the demographic variables, children who remained in the sample had slightly younger ages in year 5 than children who were not present at year 15 ($t$(1,399) = 9.45, $p$ < .001); there was no difference between those who remained in the sample and those who were not present at year 15 on mother's age, father's age, child sex, and marital status. Regarding the early childhood indicators, children who remained in the sample had fewer missing values in self-regulation ($\chi^2$ (1) = 743.18, $p$ < .001) and neighborhood collective efficacy ($\chi^2$ (1) = 650.86, $p$ < .001) than children who were not present at year 15. Children who remained in the sample had higher missing in maternal warmth ($\chi^2$ (1) = 507.49, $p$ < .001), parental educational attainment ($\chi^2$ (1) = 264.55, $p$ < .001), and poverty status ($\chi^2$ (1) = 284.5, $p$ < .001) than children who were not present at year 15. We did not detect a missing difference in the parental impulsivity indicator between children who remained in the sample and those who were not present at year 15.

## Model selection

We tested LCA from 1 to 6 latent classes and compared them using several fit statistics, including AIC, BIC, and CAIC (see Table 3). These fit statistics help to determine how well each model fits the data. While the BIC suggested a 3-class model, the AIC continued to get smaller up to the 6-class model. However, starting at the 4-class model, the VLMR-LRT became non-significant, and in the 5-class model, one class was very small and not meaningfully different from the others. Because the fit statistics suggested models with differing numbers of latent classes, we also considered the substantive interpretability to better understand the meaningful interpretation between the 3-class and 4-class models [77]. The 3-class solution only reflected the "high," "medium," and "low" risk and protective classes, which were less informative for our research questions. In contrast, the 4-class solution offered more meaningful and nuanced distinctions. It identified classes with multidimensional protective factors and captured unique risk factors, providing richer insight into the different early-life contexts shaping adolescent risky sexual outcomes. Although the 3-class model had a slightly better BIC value, we chose the 4-class model because it provided for a more detailed and interpretable understanding of the data, particularly important for informing targeted interventions. Average posterior probabilities for most likely class membership ranged from 0.626 to 0.828 across the four classes, indicating moderate to strong classification quality (see S4 Table).

## Latent classes

Based on conditional item response probabilities of being in the "less risky/more protective" level of the category (Fig 2, S3 Table), we interpreted and labeled four classes based on their distinct patterns of risk and protective factors. The Multidimensional Protective Class ($n$ = 706, 21.5%) had the highest probability of experiencing multiple types of protective factors (e.g., children with high self-regulation, high maternal warmth, high neighborhood collective efficacy, high parental education, parents in the non-poor group, parents with low impulsivity), with all item response probabilities over 0.6. Individuals in the Socioeconomic Status (SES) Risk Class ($n$ = 1,943, 59.2%) had a roughly even probability of endorsing either high or low levels in children's self-regulation, maternal warmth, and neighborhood collective efficacy but were marked by a higher probability of having a parent with high school or below education attainment, and parents categorized as poor. This class

**Table 1. Descriptive Statistics for Latent Class Analysis Indicators and Adolescents' Sexual Outcomes.**

| Variable | N | Frequency |
|---|---|---|
| *Early Childhood Risk and Protective Factors* | | |
| Children Self-regulation (Y5) | | |
| Low | 1,085 | 44.6% |
| High | 1,350 | 55.4% |
| Maternal Warmth (Y5) | | |
| Low | 806 | 44.4% |
| High | 1,008 | 55.6% |
| Neighborhood Efficacy (Y5) | | |
| Low | 1,152 | 42% |
| High | 1,593 | 58% |
| Mother Impulsivity (Y5) | | |
| Low | 1,595 | 49.9% |
| High | 1,600 | 50.1% |
| Father Impulsivity (Y5) | | |
| Low | 1,334 | 54.8% |
| High | 1,101 | 45.2% |
| Mother's Educational Attainment (Y5) | | |
| High school or below | 1,604 | 46.6% |
| College and higher | 1,599 | 46.4% |
| Father's Educational Attainment (Y5) | | |
| High school or below | 1,294 | 37.6% |
| College and higher | 1,067 | 31.0% |
| Mother's poverty status (Y5) | | |
| Poor Group | 2,118 | 61.5% |
| Non-poor Group | 1,088 | 31.6% |
| Father's poverty status (Y5) | | |
| Poor Group | 1,169 | 33.9% |
| Non-poor Group | 1,277 | 37.1% |
| *Adolescent Sexual Outcomes* | | |
| Sexual initiation (Y15) | | |
| Yes | 719 | 20.9% |
| No | 2,677 | 77.7% |
| Early sexual initiation | | |
| Sex before age 16 | 170 | 4.9% |
| Sex after age 16 (includes age 16) | 3,253 | 94.5% |
| Sex without a condom at first intercourse | | |
| Yes, unprotected | 117 | 3.4% |
| No, protected | 3,309 | 96.1% |
| Multiple sexual partners | | |
| Yes, sex with 2 or more people | 380 | 11.0% |
| No | 3,017 | 87.6% |

*Note*. N reflects the total number of observations for the focused variables. Percentages for categorical variables in the analytic sample. Y5 = Year 5. Y15 = Year 15.

**Table 2. Pearson Correlation table among variables in Latent Class Analysis (N = 3,281).**

| | | 1 | 2 | 3 | 4 | 5 | 6 | 7 | 8 | 9 | 10 | 11 | 12 | 13 | 14 | 15 | 16 | 17 | 18 | 19 |
|---|---|---|---|---|---|---|---|---|---|---|---|---|---|---|---|---|---|---|---|---|
| 1 | Children SR (Y5) | 1 | | | | | | | | | | | | | | | | | | |
| 2 | Maternal Warmth (Y5) | .10*** | 1 | | | | | | | | | | | | | | | | | |
| 3 | Neighborhood CE (Y5) | .15*** | .14*** | 1 | | | | | | | | | | | | | | | | |
| 4 | Mother Impulsivity (Y5) | .16*** | .05* | .15*** | 1 | | | | | | | | | | | | | | | |
| 5 | Father Impulsivity (Y5) | .08*** | .05* | .05* | .07*** | 1 | | | | | | | | | | | | | | |
| 6 | Mother Education (Y5) | .12*** | .13*** | .11*** | .11*** | .14*** | 1 | | | | | | | | | | | | | |
| 7 | Father Education (Y5) | .13*** | .09*** | .16*** | .12*** | .20*** | .32*** | 1 | | | | | | | | | | | | |
| 8 | Mother Poverty Status (Y5) | .13*** | .17*** | .22*** | .13*** | .15*** | .33*** | .33*** | 1 | | | | | | | | | | | |
| 9 | Father Poverty Status (Y5) | .11*** | .13*** | .14*** | .11*** | .15*** | .27*** | .34*** | .48*** | 1 | | | | | | | | | | |
| 10 | Sex Initiation (Y15) | −.09*** | −.07** | −.05* | −.06*** | −.06** | −.12*** | −.10*** | −.14*** | −.08*** | 1 | | | | | | | | | |
| 11 | Early Sex Initiation (Y15) | −.08*** | −.02 | −.04 | −.04* | −.03 | −.04* | −.04 | −.08*** | −.02 | .45*** | 1 | | | | | | | | |
| 12 | Unprotected Sex FI (Y15) | −.08*** | −.02 | −.01 | −.01 | −.03 | −.05** | −.03 | −.06*** | −.04 | .37*** | .20*** | 1 | | | | | | | |
| 13 | Multiple Sexual Partners (Y15) | −.08*** | −.04 | −.04* | −.05** | −.04 | −.08*** | −.07*** | −.09*** | −.07*** | .71*** | .44*** | .23*** | 1 | | | | | | |
| 14 | Child Sex | .09*** | .06** | .01 | .01 | 0 | .01 | 0 | −.01 | −.02 | −.18*** | −.16*** | −.05** | −.19*** | 1 | | | | | |
| 15 | Child Age (Y5) | .03 | −.02 | −.02 | −.04** | −.03 | −.07*** | −.06** | −.07*** | −.05** | .10*** | .04* | .02 | .08*** | .03 | 1 | | | | |
| 16 | Mother Age (Y5) | .14*** | .12*** | .10*** | .07*** | .15*** | .19*** | .22*** | .24*** | .22*** | −.11*** | −.05** | −.04* | −.08*** | .02 | −.01 | 1 | | | |

*(Continued)*

**Table 2.** (Continued)

| | | 1 | 2 | 3 | 4 | 5 | 6 | 7 | 8 | 9 | 10 | 11 | 12 | 13 | 14 | 15 | 16 | 17 | 18 | 19 |
|---|---|---|---|---|---|---|---|---|---|---|---|---|---|---|---|---|---|---|---|---|
| 17 | Father Age (Y5) | .13*** | .10*** | .08*** | .05** | .13*** | .15*** | .20*** | .21*** | .14*** | −.06** | −.04* | −.04* | −.04* | .03 | .01 | .74*** | 1 | | |
| 18 | Mother Race | 0 | −.04 | −.13*** | −.06*** | −.11*** | −.14*** | −.13*** | −.19*** | −.15*** | 0 | .01 | .02 | .01 | −.01 | .12*** | −.08*** | −.10*** | 1 | |
| 19 | Marital Status | .12*** | .15*** | .16*** | .11*** | .15*** | .24*** | .31*** | .37*** | .28*** | −.14*** | −.08*** | −.06*** | −.12*** | −.01 | −.05** | .40*** | .34*** | −.12*** | 1 |

*Note*. Reported values are Pearson correlation coefficients (*r*). SR = Self-regulation, CE = Collective Efficacy, FI = First Intercourse. Y5 = year 5. Y15 = year 15.

* *p* < .05. ** *p* < .01. *** *p* < .001.

**Table 3. Fit statistics for 1 to 6 Class Solution.**

| No. of Classes | −2LL | AIC | BIC | CAIC | VLMR-LRT | VLMR-LRT *p*-value | Entropy |
|---|---|---|---|---|---|---|---|
| 1-Class | 32540.84 | 32556.84 | 32613.69 | 32622.69 | – | – | 1 |
| 2-Class | 30755.94 | 30793.94 | 30909.76 | 30928.76 | 1763.12 | 0 | .68 |
| 3-Class | 30686.68 | 30744.68 | 30921.47 | 30950.47 | 68.41 | .01 | .68 |
| **4-Class** | **30623.70** | **30701.70** | **30939.44** | **30978.44** | **62.21** | **.13** | **.62** |
| 5-Class | 30582.50 | 30680.50 | 30979.20 | 31028.20 | 40.71 | .19 | .54 |
| 6-Class | 30545.01 | 30663.01 | 31022.66 | 31081.66 | 37.03 | .08 | .54 |

*Note*. AIC = Akaike Information Criteria. BIC = Bayesian Information criterion. CAIC = Consistent Akaike Information Criteria. VLMR-LRT = Vuong-Lo-Mendell-Rubin Adjusted Likelihood Ratio Test. Bold indicates the selected model.

was also marked by a relatively high probability of fathers with high impulsivity. The Neighborhood Risk Class (*n* = 239, 7.3%) had a roughly even probability of endorsing items in either high or low levels in children's self-regulation, maternal warmth, mother-reported poverty status, and father's educational attainment, with a higher probability of having a more highly educated mother, a non-poor father and a mother with high impulsivity; however, these children were most likely to live in neighborhoods with low collective efficacy. The Low Father Education Class (*n* = 393, 12%) was marked by a higher probability of higher children's self-regulation, higher maternal warmth, non-poor mother, and a mother with low impulsivity; however, individuals in this class were likely to have fathers with high school or lower education.

## Demographics correlates of latent classes membership

Next, we used multinomial logistic regression to examine how demographic characteristics were associated with class membership (Table 4). There were multiple significant differences between classes in terms of the mother's race. Children with White mothers had higher odds of membership in the Multidimensional Protective class (OR = 2.30; *p* = .002) than the Low Father Education class compared to children with Black mothers; however, children with Latinx mothers had lesser odds of membership in the Multidimensional Protective Class (OR =.46; *p* = .04) compared to children with Black mothers. Children with Latinx mothers had higher odds of membership in the SES Risk Class (OR = 2.48; *p* = .02) than the Neighborhood Risk Class compared to children with Black mothers; however, children with Other/Mixed race mothers had lesser odds of being in the SES Risk Class (OR =.26; *p* = .01) than the Neighborhood Risk Class compared to children with Black mothers. Additionally, children with married and older mothers had higher odds of being in the Multidimensional Protective class compared to the other three classes. Compared to the SES Risk and Neighborhood Risk classes, younger children have lower odds of membership in the Multidimensional Protective Class.

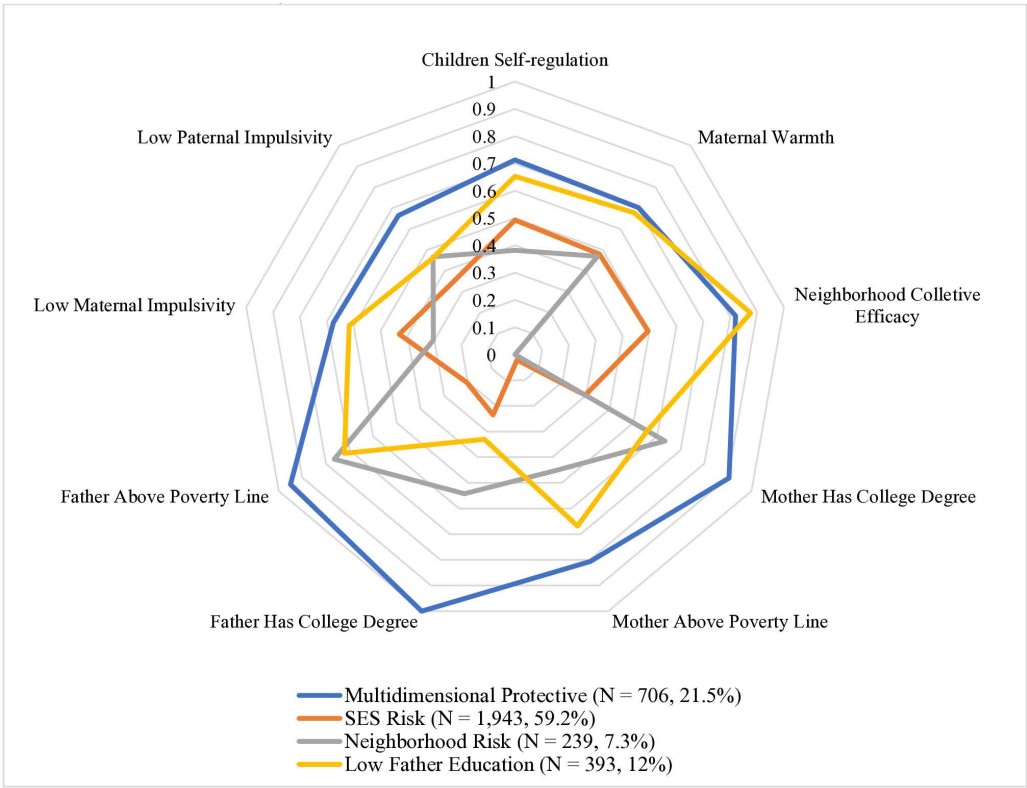

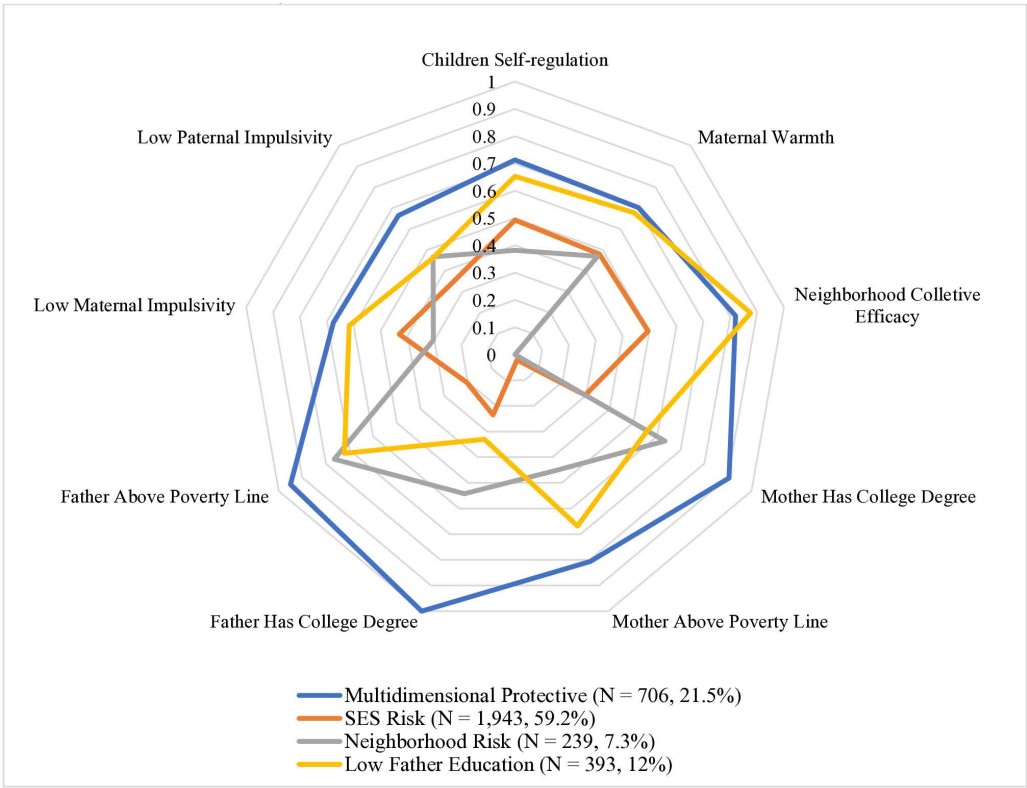 is the radar chart legend positioned below the figure:

- Multidimensional Protective (N = 706, 21.5%)
- SES Risk (N = 1,943, 59.2%)
- Neighborhood Risk (N = 239, 7.3%)
- Low Father Education (N = 393, 12%)

*Note.* Item response probability represents the prevalence of individuals in a given class selecting the less risky/more protective level of the variable. SES = Socioeconomic status.

**Fig 2. Item Response Probabilities and Group Membership Prevalence for Latent Classes of Protective and Risk Factors at Age 5 (N = 3,281).**

**Table 4. Odds Ratios, Confidence Interval, and p-values of Demographic Variables on Protective and Risky Indicators.**

| | Socioeconomic Status RiskSESR | | Neighborhood RiskNR | | Low Father EducationLFE | | Multidimensional Protective MP (Reference) | All Significant (p<.05) Class Differences |
|---|---|---|---|---|---|---|---|---|
| | Odds Ratio | 95% CI | Odds Ratio | 95% CI | Odds Ratio | 95% CI | | |
| Child Sex | 1.00 | [0.72, 1.40] | 1.19 | [0.69, 2.03] | 1.04 | [0.66, 1.64] | 1 | |
| Child Age | 1.07* | [1.01, 1.14] | 1.10* | [1.00, 1.20] | 1.04 | [0.96, 1.12] | 1 | MP<SESR, NR |
| Mother Age | 0.89*** | [0.85, 0.93] | 0.92* | [0.86, 0.99] | 0.89*** | [0.84, 0.95] | 1 | MP>SESR, NR, LFE |
| Father Age | 1.02 | [0.98, 1.06] | 1.02 | [0.96, 1.09] | 1.02 | [0.97, 1.07] | 1 | |
| Marital Status | 0.11*** | [0.08, 0.16] | 0.17*** | [0.08, 0.38] | 0.24*** | [0.14, 0.42] | 1 | MP>SESR, NR, LFE; LFE>SESR |
| Same-sex Attraction | 1.92 | [0.92, 4.02] | 0.86 | [0.20, 3.62] | 1.87 | [0.76, 4.61] | 1 | |
| Non-Hispanic White | 0.18*** | [0.12, 0.27] | 0.27** | [0.13, 0.59] | 0.44** | [0.26, 0.75] | 1 | MP>SESR, NR, LFE; LFE>SESR |
| Hispanic | 1.88* | [1.06, 3.34] | 0.76 | [0.29, 1.98] | 2.18* | [1.04, 4.56] | 1 | SESR, LFE>MP; SESR>NR; NR<LFE |
| Other/Mixed race | 0.33* | [0.14, 0.77] | 1.29 | [0.45, 3.68] | 0.41 | [0.12, 1.48] | 1 | SESR<MP, NR |

*Note.* The values represent the odds ratio with the 95% confidence interval (CI). The reference class is the Multidimensional Protective class. The reference categories for child sex is boy, for marital status is single, for the racial group is the Black/African American. All significant comparisons are included in the All Significant Class Differences column.

* $p<.05$. ** $p<.01$. *** $p<.001$.

## Predicting adolescent sexual behaviors

We weighted participants based on their probability of latent class membership and conducted weighted logistic regressions using the BCH approach to examine the association between latent class membership and adolescents' sexual behaviors at year 15, controlling for demographic covariates. Results revealed that latent class was a significant predictor for all outcomes at year 15 (sexual initiation: $\chi^2 = 96.43$, $p < .001$; early sexual initiation: $\chi^2 = 18.15$, $p < .001$; sex without a condom at first intercourse: $\chi^2 = 20.27$, $p < .001$; multiple sexual partners: $\chi^2 = 54.88$, $p < .001$), suggesting prevalence rates differed significantly across latent classes. Subsequently, we conducted paired comparisons between latent classes to investigate the specific differences in sexual behaviors (Table 5). The SES Risk class had the highest prevalence of sexual initiation (27%), early sexual initiation (7%), sex without a condom at first intercourse (5%), and multiple sexual partners (14%), and was significantly higher than the Multidimensional Protective class. The Multidimensional Protective class had a significantly lower prevalence of sexual initiation (7%) compared to the Neighborhood Risk class (18%) and Low Father Education class (22%). The SES Risk class had a higher prevalence of having sex without a condom at first intercourse (5%) than the Neighborhood Risk class (1%). The Low Father Education class (12%) had a higher prevalence of having multiple sexual partners than the Multidimensional Protective class (3%).

## Discussion

Adolescents' risky sexual behaviors are a serious public health concern and are related to adverse health outcomes in later life [4,5]. This study provides a comprehensive exploration of the patterns of early childhood risk and protective factors within a racially diverse sample and examines the long-term association with adolescents' sexual outcomes. As predicted, we uncovered distinct classes marked by specific risk factors, such as SES risk, neighborhood risk, and low father education, while also uncovering a class with lower risk and multiple protective factors. The multidimensional protective class has the lowest prevalence of sexual outcomes during adolescence compared to the other three classes. Conversely, the SES risk class is linked to the higher risk of early sexual initiation, sex without a condom at first intercourse, and having multiple sexual partners. Identifying early childhood risk and protective factors and their long-term impact on adolescents' risky sexual behaviors can support the implementation of targeted and tailored intervention programs for adolescents with specific risks.

### Classes of early risk and protective factors

In line with our initial hypothesis, we identified four unique latent classes that are characterized by domain-specific risks, and also a class with multiple protective factors. This further extends and aligns with the social-ecological framework by examining the co-occurrence of interconnected and dynamic social contexts across multiple domains in early childhood.

Table 5. Probabilities of Early Childhood Membership on Sexual Behavior Outcomes in Adolescence.

| Variables | Class 1: Multidimensional Protective[MP] Prevalence | Class 2: SES Risk[SESR] Prevalence | Class 3: Neighborhood Risk[NR] Prevalence | Class 4: Low Father Education[LFE] Prevalence | Pairwise Comparisons |
|---|---|---|---|---|---|
| Sexual Initiation | .07 | .27 | .18 | .22 | **MP<SESR\*\*\*, NR\*, LFE\*\*\*** |
| Early Sexual Initiation | .02 | .07 | .04 | .04 | **MP<SESR\*\*\*** |
| Sex Without a Condom at First Intercourse | .01 | .05 | .01 | .03 | **MP<SESR\*\*\* SESR>NR\*** |
| Multiple Sexual Partners | .03 | .14 | .09 | .12 | **MP<SESR\*\*\*, LFE\*\*** |

*Note.* All significant comparisons are included in the Pairwise Comparisons column. SES = Socioeconomic status.

\* $p < .05$. \*\* $p < .01$. \*\*\* $p < .001$.

The majority of participants were in the SES Risk class. Given that the FFCWS purposefully oversampled children with non-married parents from low-income backgrounds, this class emerged as the most prevalent risk factor in our current sample. We also found that one-fourth of participants were in the Multidimensional Protective class, the class marked by a high probability of experiencing all protective factors and a low probability of exposure to risk factors, suggesting the co-occurrence among these factors. Furthermore, two smaller classes with domain-specific risks were identified, including the Neighborhood Risk class, characterized by low neighborhood collective efficacy, and the Low Father Education class, marked by fathers with lower educational levels. These findings suggest that children in our sample experienced diverse early-life contexts in early childhood, with some of them residing in environments marked by stressful and specific risk challenges.

### Demographic predictors of class membership

We also found that class membership differed by diverse demographic characteristics in line with our second hypothesis. Specifically, being a racial/ethnic minority emerges as a significant factor associated with being in latent classes marked by risk factors. Children with White mothers compared to Black mothers have a higher probability of belonging to the Multidimensional Protective class than the Low Father Education class. Similarly, children with Latinx mothers compared to Black mothers have a higher probability of belonging to the SES Risk class than the Neighborhood Risk class. These findings were consistent with prior literature suggesting that Black individuals are more likely to be in a class marked by neighborhood risk [61]. In addition, marital status was also a significant predictor. Compared to the three other risk classes, children in the Multidimensional Protective class had a higher probability of having a mother who was married at the child's birth. This result is consistent with prior literature suggesting that married mothers are less likely to be in poverty or live in disadvantaged neighborhoods [78]. Thus, our findings are consistent with prior research and highlight the ways in which adolescents from different demographic backgrounds may be exposed to different types of contextual risk.

### Early risk and protective class memberships predict adolescent sexual behaviors

We found several associations between class membership and adolescents' sexual behaviors. Adolescents in the Multidimensional Protective class demonstrated a lower prevalence of sexual initiation compared to the other three risk classes. The Multidimensional Protective class also had a lower likelihood of early sexual initiation, sex without a condom at first intercourse, and multiple sexual partners compared to the SES Risk class. This aligns with our third hypothesis and suggests that adolescents within the Multidimensional Protective class may experience a more supportive and nurturing early life environment. Such environments appear to act as a safeguard, reducing the vulnerability to engage in risky sexual behaviors. The co-occurrence of parental warmth, children's self-regulation, and lower parental impulsivity, better SES, could serve as a buffer against early or unprotected sexual activity. This also aligns with the risk and protective framework [15], highlighting that protective factors operating across diverse domains collectively contribute to mitigating the negative impacts of the risk factors and are impactful in protecting against sexual behaviors.

In contrast, the SES Risk class had the highest prevalence of early sexual initiation and was significantly different from the Multidimensional Protective class. This finding is consistent with life history theory [17], suggesting that individuals living in impoverished early life environments perceive consistent economic hardship and limited resources and thus may adopt fast reproductive strategies as an adaptive trade-off to ensure reproductive success. This adaptation manifests in behaviors such as engaging in early sex during adolescence, which allows individuals to seize reproductive opportunities in an environment where uncertainties about resource availability persist. This finding also extends existing literature, which found an association between family SES during adolescence and adolescent sexual behaviors [43,44] by showing that early SES also impacts adolescents' later sexual health. The co-occurrence of lower parental education and limited family income in this class may contribute to heightened stress reactivity [79], fostering conditions conducive to early

sexual behavior. It also aligns with broader developmental literature, highlighting that the early socio-economic context and an individual's sexual health development are particularly sensitive and vulnerable to the early childhood environments [80,81]. Our findings suggest that early SES is a specific risk factor and provide a more comprehensive understanding of the intricate relationships between socio-environmental factors and reproductive behaviors across the life course. This underscores the importance of the implementation of early intervention programs to address socioeconomic disparities in early childhood as a pivotal strategy for promoting positive sexual health outcomes in adolescence.

Furthermore, the Low Father Education class had a higher prevalence of having multiple sexual partners than the Multidimensional Protective class. A potential explanation is that there may exist differences in the role of fathers with lower levels of educational attainment in the household. For instance, fathers in this specific class might work multiple jobs with unstable shifts due to limited access to higher occupation positions constrained by diploma or skill requirements [82]. Due to financial constraints and work burdens, fathers may experience parenting strain, particularly when their employment is not flexible enough to take care of children, potentially impacting children's well-being [83]. Additionally, fathers in this class may have reduced opportunities for communication or monitoring due to work-related constraints, which could limit their availability and engagement with children. This reduced involvement may affect the messages children receive about sexual behaviors. In the absence of consistent paternal guidance and emotional support, adolescents may engage in multiple sexual partnerships as a way to seek connection, assert autonomy, or cope with emotional needs that are unmet in their family environment. Compared to the Multidimensional Protective class, children in the Low Father Education class had a greater likelihood of having single mothers, which may also influence adolescents' sexual behaviors. The finding echoes research, such that the single-parent household had a higher prevalence of sexual behaviors [45]. As such, according to life history theory, individuals may adjust their reproductive strategies based on environmental cues and available resources. The potential stressors associated with lower educational attainment may be related to reduced involvement. This, in turn, could influence the sexual behavior of adolescents, potentially leading to a higher likelihood of engaging in multiple sexual partnerships. These structural and relational dynamics together reflect a constellation of risk that may undermine consistent parental guidance and increase adolescents' susceptibility to risk-seeking behaviors. This suggests that interventions for this class may need to include accessible father engagement programs and supports for co-parenting structures in lower-resource households, particularly those addressing time constraints and involvement in adolescent development.

The SES Risk class had a higher prevalence of engaging in sex without a condom at first intercourse compared to the neighborhood risk class. This heightened vulnerability among adolescents in the SES Risk class may reflect the compounding effects of multiple family-level disadvantages, such as economic hardship and low parental education. A potential explanation is that families with lower SES may experience heightened stressors, such as unstable employment, which may reduce parental involvement [84,85]. Adolescents from socioeconomically disadvantaged backgrounds are more prone to substance use [86], which may be associated with riskier sexual behavior [87]. Youths growing up in lower SES households may also hold ambivalent attitudes towards pregnancy, which could manifest as a fluctuation between a desire to avoid pregnancy and a tendency to romanticize it as a desirable escape from their current lives in which other opportunities are limited and a source of meaning and identity [88,89,90]. This ambivalence may contribute to adolescents using condoms less consistently, including not using a condom at first intercourse. In contrast, although adolescents in the neighborhood risk class did not exhibit the highest levels of risky sexual behavior, the early exposure to low neighborhood collective efficacy may still pose long-term developmental challenges. The lack of social cohesion and informal social control may reduce exposure to prosocial norms and opportunities. While this class may not be marked by direct family-level disadvantage, the broader community environment could still shape developmental trajectories in subtler but meaningful ways. Interventions that target neighborhood-level resources, such as expanding safe recreational spaces, strengthening community mentorship networks, and implementing violence prevention initiatives, may help foster collective efficacy and provide youth with protective alternatives to risk behaviors.

## Implications

The current findings emphasize the importance of early, multi-tiered prevention and intervention programs to reduce adolescent sexual risk behaviors. First, adolescents in the Multidimensional Protective class had a lower likelihood of engaging in risky sexual behavior. This highlights the need to foster early protective factors across multiple levels. Specifically, developing a multifaceted approach is important, such that intervention approaches should be transcended on multiple levels [91]. For instance, family-level interventions, such as evidence-based home visiting programs and parenting education programs, could strengthen positive parenting. At the neighborhood level, community-based programs that enhance collective efficacy, such as neighborhood parenting meetups, could foster supportive social norms and reduce isolation. These intervention programs could collectively support protective factors across multiple levels.

Second, adolescents in the SES Risk class had the highest risk of engaging in early and unprotected sexual activity. This highlights the importance of integrating poverty alleviation strategies with adolescent-focused sexual health education. Economic supports, such as the Earned Income Tax Credit expansion and housing subsidies, may indirectly improve adolescent health outcomes by reducing early family economic adversity. Simultaneously, school-based sexual health programs should implement trauma-informed, culturally responsive, future-oriented sexual health programs tailored to high-SES-risk adolescents. These should move beyond basic sexual education to include identity development, goal-setting, and resilience-building components that help youth contextualize and reflect on how environmental hardship shapes their life strategies.

Third, the elevated risk of multiple sexual partnerships among adolescents in the Low Father Education class suggests a need for targeted family-based and school-based support programs. Public health initiatives could promote father-inclusive sexual health education, including group workshops, digital toolkits, or peer-led programs, that emphasize the importance of parent–child communication about sexual health, designed for fathers with lower educational attainment. In school settings, teachers and school counselors should be equipped with screening tools to detect contextual risks and refer youth to mentoring programs, peer support networks, or psychosocial services. For example, training teachers to recognize signs of disengagement or family strain and to provide referrals to relevant services can strengthen school-family partnerships and enhance adolescent well-being.

## Limitations and future directions

Although the longitudinal nature of our study is a strength, limitations should also be noted. First, the measures of adolescent sexual behaviors may not comprehensively reflect the diversity of sexual activities. For example, the measure of sex without a condom only reflected adolescents' first intercourse but did not provide a comprehensive understanding of condom usage over time. In addition, the current study did not assess other contraceptive methods, such as pills, patches, and implants, as these measures are not available in the FFCWS data. Future research should adopt a holistic measure to reflect the diversity of contraceptive methods. Second, the measure of sexual behavior in this study focused on vaginal intercourse, which may not capture the full range of sexual behaviors (e.g., oral, anal sex). This operationalization may have measurement bias, such as underestimating sexual behaviors, particularly among LGBTQ+ adolescents. While same-sex attraction was included as a covariate, the FFCWS only measured attraction, which may not reflect identity or same-sex behaviors. Given that sexual identity does not always align with sexual behavior, many LGBTQ+ adolescents may engage in vaginal intercourse [70]; current measurement may obscure sexual health disparities in marginalized populations. Future research should incorporate more inclusive and behaviorally nuanced measures to better understand early predictors of sexual behavior with larger samples of LGBTQ+ adolescents.

Third, the indicators used in the LCA were dichotomized, which may artificially categorize individuals near the mean into a high- or low-response group. This approach may reduce variability, potentially weaken the differences between the groups, and limit the ability to detect more nuanced effects. While dichotomization was used in the present study to enhance interpretability and align with prior research, future studies should consider using continuous indicators

that better reflect meaningful distinctions among individuals. Fourth, because the FFCWS sampled high percentages of low-income families that may have high residential mobility [92], it is important to measure longitudinal neighborhood information to identify the duration and the mobility of the residential status. Fifth, the adolescents' sexual behaviors were self-reported at year 15, which might have social desirability bias, such that adolescents' reported sexual behaviors could be inaccurate. A recent review highlights that social desirability could influence the accuracy of the participant's self-reported sexual behavior in the survey method, such that respondents may underreport engaging in sexual behaviors and overreport condom use [93]. These concerns may be mitigated somewhat by the use of advanced technology, such as computer-assisted or indirect questions, to provide greater privacy and anonymity that may reduce social desirability bias [93]. Sixth, the FFCW study oversampled low-income households, and subsequently, the largest class was marked by SES risk. This class structure may be replicated by other urban high-risk samples that comprise single households and racial minorities similar to the current sample's characteristics. However, the findings may not be generalized to nationally representative samples or lower-risk samples, which may have a different class structure. For example, in a nationally representative sample, the multidimensional protective class could be more prevalent, and the SES risk class or low father education class may be less prevalent. Seventh, there is a potential for interviewer bias, particularly for variables based on maternal self-report. Although measures are drawn from validated instruments, subjective responses may still introduce measurement error. Such misclassification could attenuate or distort associations between early contextual factors and adolescent sexual outcomes, underscoring the importance of incorporating multi-method and multi-informant approaches in future research to strengthen the validity of findings. Furthermore, a strength of this study is the use of a racially and ethnically diverse sample that includes a large proportion of children from socioeconomically disadvantaged and single-parent households. This allowed for a nuanced examination of early contextual risks often underrepresented in national studies. However, this may limit the generalizability of the findings to the broader U.S. population. Specifically, the distribution of latent class membership and the prevalence of adolescent sexual behaviors may not reflect national estimates. Future research should replicate these findings in more representative samples to assess generalizability. Lastly, while this study captured a range of early risk and protective factors across family and neighborhood domains, several influential contextual variables were not measured. We did not account for peer influences, media exposure, or school climate, factors that have been linked to adolescent sexual health [94]. Future research should incorporate these contextual factors to provide a more comprehensive understanding of the multiple, interacting influences on adolescent sexual development.

## Conclusion

This study provides unique contributions to understanding adolescent sexual behaviors from early childhood social-ecological contexts. Using a person-centered approach, we discovered four distinct classes of multidimensional risk and protective factors in early childhood. These meaningful classes and their long-term association with adolescent sexual behaviors suggest unique patterns of early risk factors, such as SES risk, neighborhood risk, and low father education. These results emphasized the importance of understanding the underlying processes from early childhood and how they impacted long-term development into adolescence. Our study informs future research to explore how early childhood contextual SES risk can be intertwined with the complexity of adolescent risk behaviors and the significance of early intervention programs to support adolescents' development.

## Supporting information

**S1 Table. Descriptive Statistics for Original Continuous Variables.**
(DOCX)

**S2 Table. Preliminary Analysis of Latent Profile Analysis with Continuous Variables (4-profile model).**
(DOCX)

**S3 Table. Latent Class Prevalence and Item-Response Probabilities for Four-Class Model of Protective and Risky Indicators.**
(DOCX)

**S4 Table. Average Latent Class Probabilities for the Four-Class Model of Protective and Risky Indicators.**
(DOCX)

**S1 File: Mplus and R coding.**
(DOCX)

## Author contributions

**Conceptualization:** Qingyang Liu.

**Data curation:** Qingyang Liu.

**Formal analysis:** Qingyang Liu.

**Funding acquisition:** Sara A. Vasilenko, Xiafei Wang.

**Methodology:** Qingyang Liu, Sara A. Vasilenko.

**Project administration:** Qingyang Liu, Sara A. Vasilenko.

**Software:** Qingyang Liu.

**Supervision:** Rachel A. Razza.

**Writing – original draft:** Qingyang Liu, Sara A. Vasilenko.

**Writing – review & editing:** Qingyang Liu, Sara A. Vasilenko, Xiafei Wang, Rachel A. Razza.

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
