## [Decision Letter · Decision Letter 0]

9 Jun 2025

PONE-D-25-04608Early childhood risk and protective factors and their association with adolescent sexual behaviors: A Latent Class AnalysisPLOS ONE

Dear Dr. Liu,

Thank you for submitting your manuscript to PLOS ONE. After careful consideration, we feel that it has merit but does not fully meet PLOS ONE’s publication criteria as it currently stands. Therefore, we invite you to submit a revised version of the manuscript that addresses the points raised during the review process. I strongly recommend that you carefully consider all the reviewers' comments and make an effort to address their concerns. While some reviewers requested brevity, others sought more detailed explanations of your process. Thus, I suggest you strive for clarity and conciseness in your writing. Please submit your revised manuscript by Jul 24 2025 11:59PM. If you will need more time than this to complete your revisions, please reply to this message or contact the journal office at plosone@plos.org . Please include the following items when submitting your revised manuscript:

We look forward to receiving your revised manuscript.

Kind regards,

Ayodeji Babatunde Oginni

Academic Editor

PLOS ONE

Journal Requirements:

[Funding: This research was supported by grant R03 HD096101 from the Eunice Kennedy Shriver National Institute of Child Health and Human Development and a David B. Falk College of Sport and Human Dynamics Tenure-track Assistant Professor Research Seed Grant to Sara Vasilenko and Xiafei Wang. The content is solely the responsibility of the authors and does not necessarily represent the official views of NICHD or the National Institutes of Health.].

4. Thank you for stating the following in your manuscript:

[Acknowledgment: Funding for the Future of Families and Child Wellbeing Study was provided by the Eunice Kennedy Shriver National Institute of Child Health and Human Development (NICHD) of the National Institutes of Health under award numbers R01HD36916, R01HD39135, and R01HD40421, as well as a consortium of private foundations. No direct funding was received from the sources mentioned above.

Funding: This research was supported by grant R03 HD096101 from the Eunice Kennedy Shriver National Institute of Child Health and Human Development and a David B. Falk College of Sport and Human Dynamics Tenure-track Assistant Professor Research Seed Grant to Sara Vasilenko and Xiafei Wang. The content is solely the responsibility of the authors and does not necessarily represent the official views of NICHD or the National Institutes of Health.]

[Funding: This research was supported by grant R03 HD096101 from the Eunice Kennedy Shriver National Institute of Child Health and Human Development and a David B. Falk College of Sport and Human Dynamics Tenure-track Assistant Professor Research Seed Grant to Sara Vasilenko and Xiafei Wang. The content is solely the responsibility of the authors and does not necessarily represent the official views of NICHD or the National Institutes of Health.]

Reviewers' comments:

Reviewer's Responses to Questions

**Comments to the Author**

1. Is the manuscript technically sound, and do the data support the conclusions?

Reviewer #1: Yes

Reviewer #2: Yes

Reviewer #3: Yes

Reviewer #4: Yes

Reviewer #5: Yes

Reviewer #6: Yes

Reviewer #7: Yes

2. Has the statistical analysis been performed appropriately and rigorously? 

Reviewer #1: Yes

Reviewer #2: Yes

Reviewer #3: Yes

Reviewer #4: No

Reviewer #5: Yes

Reviewer #6: Yes

Reviewer #7: Yes

3. Have the authors made all data underlying the findings in their manuscript fully available?

Reviewer #1: Yes

Reviewer #2: Yes

Reviewer #3: Yes

Reviewer #4: No

Reviewer #5: Yes

Reviewer #6: Yes

Reviewer #7: Yes

4. Is the manuscript presented in an intelligible fashion and written in standard English?

Reviewer #1: Yes

Reviewer #2: Yes

Reviewer #3: Yes

Reviewer #4: No

Reviewer #5: Yes

Reviewer #6: Yes

Reviewer #7: Yes

5. Review Comments to the Author

Reviewer #1: Major Comments

Dichotomization of Variables

The decision to dichotomize continuous and ordinal variables should be more thoroughly justified. While interpretability is important, dichotomization may reduce variability, obscure nuanced associations, and introduce bias. A more transparent rationale (e.g., via sensitivity analyses or reference to cut points used in prior literature) and acknowledgment of this limitation in the discussion would be valuable.

Inclusion of Sexual Minority Adolescents

The manuscript includes adolescents with same-sex attraction but relies heavily on vaginal intercourse-based indicators of sexual behavior. This may inadvertently exclude or misrepresent sexual risk among LGBTQ+ youth. The authors should discuss the potential for measurement bias and consider the implications for interpreting sexual health disparities in marginalized populations.

Interpretation of Latent Classes

The latent classes are well described, but more narrative synthesis around how specific combinations of risk/protective factors in each class contribute to the observed adolescent outcomes would improve interpretability and relevance for intervention design.

Policy and Practice Implications

The discussion could be expanded to better articulate how these findings might inform early childhood policy, parenting interventions, or school-based programs targeting sexual health—especially for youth from low-SES or high-risk backgrounds. Consider integrating concrete recommendations for prevention or screening strategies.

Figure Clarity

Figure 1 (item response probabilities by latent class) lacks sufficient clarity. Enhancing color contrast, labeling axes clearly, and adding a legend or direct annotations would make the figure more reader-friendly and informative.

Minor Comments

Terminology Consistency: Ensure consistent use of terms such as “risk factors,” “protective factors,” “contexts,” and “profiles” throughout the text.

Statistical Significance Presentation: In Table 5, consider bolding or asterisking statistically significant pairwise comparisons for improved readability.

Ethical Considerations: Though the study uses publicly available data, a brief note confirming IRB approval for the original study and informed consent would clarify ethical safeguards.

Recent Literature: Including more recent citations (post-2020) related to adolescent sexual behavior, especially in diverse and marginalized populations, would enhance the manuscript’s currency and relevance.

Supplementary Materials: If available, providing the R or Mplus code for the LCA and BCH analysis as supplementary material would increase transparency and replicability.

Reviewer #2: It's a well-written manuscript. Every part is explained in detail. However, it takes very long to capture the whole point. Especially the theoretical background is very long. It can use some shortening.

Reviewer #3: Reviewer Comments on Manuscript PONE-D-25-04608

Title: Early childhood risk and protective factors and their association with adolescent sexual behaviors: A Latent Class Analysis

Recommendation: Minor Revision

General Assessment

This manuscript offers a significant contribution to the literature on adolescent sexual behavior by examining how early childhood risk and protective factors, across ecological domains, shape sexual outcomes in adolescence. The use of a person-centered approach via Latent Class Analysis (LCA), in a diverse, longitudinal dataset, is both timely and methodologically appropriate. The theoretical framing—drawing from life history theory, the social-ecological model, and risk and resilience perspectives—adds conceptual richness to the study.

The manuscript is well-written, logically structured, and thorough. However, certain areas require clarification or expansion, particularly around methodological choices and the interpretation of findings. With minor revisions, this work will be suitable for publication.

Major Comments

Justification for Dichotomization of Indicators

While the decision to dichotomize continuous indicators is acknowledged and justified by prior literature, it inherently reduces variability and interpretability. I recommend elaborating on this trade-off in the limitations or methods section, and possibly including supplementary information (e.g., original distributions) to reassure readers of the robustness of class distinctions.

Interpretation of Smaller Classes

The discussion primarily focuses on the SES Risk and Multidimensional Protective classes. However, the Neighborhood Risk and Low Father Education classes offer valuable insights. A more in-depth interpretation of these profiles and their implications for targeted interventions would strengthen the discussion.

Policy and Practice Implications

The conclusion briefly mentions the need for early interventions. To enhance its impact, I suggest identifying specific types of interventions (e.g., neighborhood-level programs, parenting education, school-based support for children from low SES backgrounds). This would make the findings more actionable for practitioners and policymakers.

Model Selection Clarification

Although the manuscript explains why the 4-class model was chosen over the 3-class alternative, readers less familiar with LCA might benefit from a more explicit explanation of why interpretability was prioritized over fit statistics (e.g., AIC/BIC).

Minor Comments

Title & Abstract: Clear and appropriate. However, the abstract conclusion could be more actionable—consider specifying the nature of "targeted interventions."

Theoretical Framework: Strong and well-integrated, but could be tightened for brevity, especially in overlapping explanations of life history theory.

Figures & Tables: Ensure that Figure 1 is clearly labeled and legible, especially for readers interpreting class probabilities.

Limitations: Consider mentioning unmeasured contextual variables (e.g., peer influences, media exposure) that may also shape adolescent sexual behavior.

Conclusion

This is a well-executed and theoretically sound manuscript that aligns with PLOS ONE's interdisciplinary scope. It has the potential to inform early preventive strategies in adolescent sexual health and development. I recommend minor revisions aimed at expanding interpretation, clarifying methodological decisions, and enhancing the practical relevance of the findings.

Reviewer #4: Review notes

1- The text of the article and its number of pages is too long.

2- The introduction of the text is too long.

3- The validity and reliability of the measuring instrument for the independent and response variables are not fully defined and explained.

4- The discovery and method of determining latent classes is not clear.

5- The statistical model of analysis in each class and in general is not specified.

6 -The criteria and statistics of the goodness-of-fit including AIC, BIC, and CAIC for which models were used?

Reviewer #5: This article makes a valuable contribution to developmental and public health research by highlighting the long-term impact of early childhood contexts on sexual health. It's well grounded in theory. The study underscores the need for early, tailored interventions and sets a foundation for future research on nuanced risk pathways. However, there are few issues that need to be addressed to improve the manuscript.

Generally, the article is too lengthy and needs to be as concise as possible for easy reading.

The introduction is robust, integrating multiple theoretical frameworks to justify the study. This provides a comprehensive lens for understanding how early childhood factors influence adolescent sexual behaviours.

Methods

The demographic characteristic description in the participants and procedures section of the methods should be moved to the descriptive statistics section of the results.

The authors should report the reliability and internal consistency of the scales used in terms of their Cronbach's alpha values.

Change the “Analytic Plan” section title to “Data Analysis” since you have already done the analysis.

Results

In Table 1, the authors should remove the percentage signs from the table. Also, since the number of observations for some of the variables is different, it will be appropriate to report the total number of observations where different for each variable, besides the variable name

For Table 2, what correlation estimator was used?

For Figure 1, the trend graph is not suitable for this data. I would entreat the authors to try a radar plot or just a simple bar plot.

Discussion

While associations are clear, the discussion could speculate more on why certain classes (Low Father Education) link to specific outcomes (multiple partners).

Reviewer #6: Thank you for putting this manuscript together. I have some suggestions that I believe that can strengthen the paper. Please address these comments.

1. “Several theoretical frameworks suggest that the contexts of early childhood shape

later sexual risk behaviors”… please provide a citation to these frameworks.

2. I like that you reviewed each frame work in the background. Can you possibly summarize these as the current state, these are voluminous to read and can be a major distraction off the import of the study.

3. Using a study with that oversample children of single parents, could this have an effect on your results (good or bad?) please state this in the strengths and limitations section

4. Method: the method section could benefit more from a flowchart than the long write up of how the analytical sample was arrived at.

5. The Demographic characteristics should not be in the method section. These should be in the results section.

6. Measures. In the method section be explicit with the measures. Be specific what measures your stud intended to describe. Go straight to the measures, this entire long write up before the measures is superfluous.

7. The first sentence under analytic plan needs to be rephrased for clarity.

8. Under the analytic plan, simply state the analyses or statistical approaches. There no need to report results under this section.

9. The model selection should be in the methods section not in the results section. Please rearrange these.

10. You listed important limitations of the study, it is important to include interviewer bias (misclassification) and its potential to bias your findings.

Reviewer #7: 1. The evidence backs up the conclusions, and the article is technically competent. Using a well-established longitudinal dataset, the study tackles a significant and current issue: the relationship between early childhood social-ecological characteristics and teenage sexual practices. The authors' research questions are well-defined, and they back up their findings with strong evidence. The results are significant, and the ramifications for public health programming and early intervention are clearly stated. The manuscript contributes significantly to the study of health and developmental behavior.

2. The statistical analysis was done correctly and thoroughly. The authors utilized the Bolck-Croon-Hagenaars (BCH) three-step method, a proven technique for connecting latent classes to external outcomes while reducing classification bias, and Latent Class Analysis (LCA) with a sizable, heterogeneous sample. Entropy values were provided, and the number of classes was determined using the model selection criteria (AIC, BIC, and VLMR-LRT). The authors defend their decision and guarantee interpretability even if dichotomizing continuous variables may result in less accuracy. All things considered, the analyses are in line with the objectives of the study and are methodologically sound.

3. All of the data supporting the conclusions has been made publicly available by the authors. They make it apparent that the information is taken from the Future of Families and Child Wellbeing Study, which is available to the public and may be obtained at https://ffcws.princeton.edu/documentation. The authors further point out that scripts to replicate findings are available upon request, promoting transparency and reproducibility, and thus conforms with open data laws.

4. The manuscript is written in standard English and is presented clearly. It conveys the goals and conclusions of the study and is clearly organized and simple to read.

I have following few minor suggestions:

1. In the Correlates of Class Membership section, the authors mentioned that six correlates were included, but seven are actually listed: child sex, child age, mother’s age, father’s age, mother’s marital status, mother’s race, and adolescents’ sexual attraction. Please revise the text to reflect the correct number.

2. Model interpretability provides justification for the 4-class model's selection. Table 3 reports entropy values, which is useful. However, the authors might think about providing average posterior probability for every class to further support the validity of class assignments.

3. The authors provide a reasonable explanation for dichotomizing continuous indicators, noting that their initial Latent Profile Analysis (LPA) did not yield substantively distinct profiles. To further strengthen this justification, the manuscript would benefit from a brief summary or table of the LPA results, showing that the continuous indicators did not meaningfully differentiate the classes (e.g., similar means across classes). Including this information, even in a supplementary table or figure, would provide empirical support for the decision to dichotomize and enhance the transparency of the analytic approach.

6. PLOS authors have the option to publish the peer review history of their article (what does this mean? ). If published, this will include your full peer review and any attached files.

**Do you want your identity to be public for this peer review?** For information about this choice, including consent withdrawal, please see our Privacy Policy .

Reviewer #1: No

Reviewer #2: **Yes: ** Afra ALKAN

Reviewer #3: **Yes: ** Juliana Aggrey

Reviewer #4: No

Reviewer #5: No

Reviewer #6: No

Reviewer #7: No

---

## [Author Response · Author response to Decision Letter 1]

4 Aug 2025

Dear Reviewers,

Thank you for the opportunity to revise our manuscript and for the thoughtful, constructive feedback provided by you and the seven reviewers. We sincerely appreciate the time and care that each of you dedicated to evaluating our work. The reviewers’ insights were instrumental in helping us clarify and strengthen the manuscript. Please see the response letter for details.

---

## [Editor Report · Decision Letter 1]

28 Aug 2025

Early childhood risk and protective factors and their association with adolescent sexual behaviors: A Latent Class Analysis

PONE-D-25-04608R1

Dear Dr. Liu,

We’re pleased to inform you that your manuscript has been judged scientifically suitable for publication and will be formally accepted for publication once it meets all outstanding technical requirements.

Within one week, you’ll receive an email detailing the required amendments. When these have been addressed, you’ll receive a formal acceptance letter, and your manuscript will be scheduled for publication.

Kind regards,

Ayodeji Babatunde Oginni

Academic Editor

PLOS ONE
---

## [Editor Report · Acceptance letter]

PONE-D-25-04608R1

PLOS ONE

Dear Dr. Liu,

I'm pleased to inform you that your manuscript has been deemed suitable for publication in PLOS ONE. Congratulations! Your manuscript is now being handed over to our production team.

Kind regards,

on behalf of

Ayodeji Babatunde Oginni

Academic Editor

PLOS ONE